# Neuropsychological Profile, Emotional/Behavioral Problems, and Parental Stress in Children with Neurodevelopmental Disorders

**DOI:** 10.3390/brainsci11050584

**Published:** 2021-04-30

**Authors:** Francesca Felicia Operto, Daniela Smirni, Chiara Scuoppo, Chiara Padovano, Valentina Vivenzio, Giuseppe Quatrosi, Marco Carotenuto, Francesco Precenzano, Grazia Maria Giovanna Pastorino

**Affiliations:** 1Child Neuropsychiatry Unit, Department of Medicine, Surgery and Dentistry, University of Salerno, 84084 Fisciano, Italy; chiara.scuoppo@gmail.com (C.S.); chiarapado@hotmail.it (C.P.); valentina.vivenzio@libero.it (V.V.); graziapastorino@gmail.com (G.M.G.P.); 2Department of Psychology, Educational Science and Human Movement, University of Palermo, 90128 Palermo, Italy; daniela.smirni@unipa.it; 3Dipartimento Promozione della Salute, Materno-Infantile, di Medicina Interna e Specialistica di Eccellenza“G. D’Alessandro”, 90127 Palermo, Italy; peppe.quatrosi@gmail.com; 4Clinic of Child and Adolescent Neuropsychiatry, Department of Mental Health and Physical and Preventive Medicine, University “Luigi Vanvitelli”, 81100 Caserta, Italy; marco.carotenuto@unicampania.it (M.C.); f.precenzano@hotmail.it (F.P.)

**Keywords:** Autism Spectrum Disorder, Specific Learning Disorder, Attention Deficit/Hyperactivity Disorder, cognitive profile, emotional/behavioral problems, parental stress

## Abstract

Background: The aim of our study was to trace a specific neuropsychological profile, to investigate emotional-behavioral problems and parental stress in children with Autism Spectrum Disorder Level 1/High functioning (ASD-HF), Specific Learning Disorders (SLD) and Attention Deficit/Hyperactivity Disorder (ADHD) disorders and to highlight similarities and differences among the three groups. Methods: We retrospectively collected the data from a total of 62 subjects with ASD-HF (*n* = 19) ADHD (*n* = 21), SLD (*n* = 22) and 20 typical development. All the participants underwent neuropsychological standardized test for the evaluation of cognitive profile (Wechsler Intelligence Scale for Children Fourth Edition—WISC-IV), behavioral and emotional problems (Child Behavior CheckList CBCL), and parental stress (Parental Stress Index Short Form—PSI-SF). The scores of the ASD-HF, ADHD, and SLD groups were compared using non-parametric statistic methods (Kruskall–Wallis H test and U Mann–Whitney for post-hoc analysis). Results: The ASD-HF group were significantly higher in all areas of the WISC-IV than the other two clinical groups. The SLD group performed significantly lower than ASD-HF in Working Memory Index. The SLD group showed lower scores on the somatic problems subscale than the other two groups. In the Difficult Child subscale of the PSI-SF, parents of ADHD children scored lower than the mothers of SLD subjects and higher than the fathers of SLD subjects. In all three groups there are specific deficiencies compared to the control group in the cognitive profile, behavioral and emotional problems, and parental stress. Conclusions: Our comparative analysis highlighted similarities and differences in three groups of children with different neurodevelopmental disorders, helping to better define cognitive, behavioral, and emotional characteristics of these children and parental stress of their parents.

## 1. Introduction

Neurodevelopmental disorders are clinical conditions that share early abnormalities in neurobiological development, and determine an impairment of personal, social, school or work functioning. Neurodevelopmental disorders may share not only an etiopathogenetic matrix [1] but also some clinical and neuropsychological characteristics [2]. Among the neurodevelopmental disorders, according to Diagnostic and Statistical Manual of Mental Disorders—Five edition (DSM-5) [1], there are Autism Spectrum Disorder (ASD), Attention Deficit/Hyperactivity Disorder (ADHD) and Specific Learning Disorders (SLD). Autism Spectrum Disorders characterized by impaired communication and socio-relational skills and restricted, repetitive, and stereotyped behaviors and interests [1]. The symptoms must be present in the early period of development and are not better explained by the intellectual disability. DSM-5 eliminated subtype characterization of ASD and introduced the term “spectrum” to emphasize the heterogeneity of the clinical features of this condition [1]. The severity of the disorder can range from very mild to severe and is divided into three levels, from 1 to 3. According to DSM-5 [1], individuals with ASD level 1, previously also called “high functioning” (ASD-HF), have intelligence and language within the limits of the norm. Several studies examined the neuropsychological profile of children with ASD-HF, showing deficit in the pragmatic aspects of language (e.g., monotonous prosody, inadequate volume, difficulty in respecting conversational shifts, limited mimicry and gestures, difficulty in understanding the double meanings and the latent meaning of the conversation). Furthermore, children with ASD-HF usually present behavior aimed at socialization, although not always functional and adequate [3]. Another neurodevelopmental disorder is Attention Deficit/Hyperactivity Disorder (ADHD), characterized by a persistent pattern of inattention and/or hyperactivity-impulsivity and [1] the worldwide prevalence in childhood population studies is about 5% [4]. To diagnose ADHD, symptoms must be present in two different contexts, develop before age 12, and have a negative impact on psychosocial functioning [1]. DSM-5 includes in neurodevelopmental disorders the Specific Learning Disorders (SLD), in which individuals have an impairment of reading, writing, and calculating skills, despite a normal intelligence. Deficient academic skills are far below the expected level for the individual’s age and involve interference with academic performance and/or daily living activities. Difficulties are not best explained by intellectual disability, impaired vision or hearing, lack of knowledge of the language of school instruction or inadequate education [1].

One of the most frequently used tests to assess intelligence in school-aged individuals is the Wechsler Intelligence Scale for Children—Fourth Edition (WISC-IV).

Some studies showed that subjects with ASD-HF score higher on some index WISC-IV while had lower in other index [3,5]. Globally, the perceptual reasoning index (PRI) is highest in ASD-HF and the working memory index (WMI) is the lowest in children with typical development [3]. According to recent studies [6,7] subjects with ADHD, have the average full-scale intelligent quotient score lower than the typically developing children and tend to perform worse on the working memory index (WMI) and on the processing speed index (PSI) than the verbal comprehension index (VCI) and the perceptual reasoning index (PRI). Many studies have demonstrated the usefulness of the WISC-IV scale [8] to evaluate the cognitive profile of subjects diagnosed with SLD [8,9]. Some authors reported that cognitive functioning of SLD subjects is different than subject with typical development [9,10,11]. The results of the Poletti study [9] showed that in these subjects the General Ability Index (GAI) usually has a higher score than the Cognitive Proficiency Index (CPI).

In literature there are few studies that compared cognitive profile of children with ASD-HF, SLD and ADHD, for example, there is a study by Craig et al. (2016) [12] in which these three categories of subjects are compared, although the cognitive level was assessed using the WISC-III. A research by Kim and Song [2] compared the ASD-HF and the ADHD subjects with Total Intelligence Quotient (TIQ) > 70, founding that Verbal Comprehension ability was significantly lower in the ASD-HF group. The ASD-HF group displayed slower processing speed, while the ADHD group exhibited poor working memory and graphomotor skills. Few studies compared cognitive profile in children with SLD and ADHD; for example, in a study by Faedda and collegues [13], the ADHD group showed lower TIQ than the SLD group, although the scores of both groups were within the mean range. Furthermore, Children with ADHD often show externalizing problems and 30–50% of individuals ADHD fulfill the criteria for Conduct Disorder (CD) or Oppositional Defiant Disorder (ODD) [14]. Biederman et al. [11] suggested that children with ADHD have poor self-regulation, low frustration tolerance, impatience, easy emotional reactions, and anger. These symptoms are directly associated with higher scores on specific Child Behavior CheckList scales (CBCL). The ADHDs may also show internalization problems, such as anxiety or depression [15].

In literature [16] studies provide evidence to support the high prevalence of behavioral and emotional problems which could result in multiple psychiatric diagnoses among children with high functioning ASD-HF. Studies which used CBCL [17,18] found significantly higher scores on the scales Withdrawn/Depressed Syndrome, Social Problems, Thinking Problems and Attention Problems in children with ASD-HF than children in a mixed clinical control group. Often there is the overlap between internalizing and externalizing problems that can be mediated with emotional dysregulation and associated neurobiological bases. Subjects with learning problems can develop externalizing and internalizing problems than subject with typical development [19]. Furthermore, there are some studies that used CBCL questionnaire, in which children with reading difficulties showed attention problems [12,20] and problems related to the social sphere [20]. Another study [19] showed more internalizing behaviors and inattention among young people with poor reading skills compared with their peers with typical reading skills during the adolescence. In the literature there are very few studies that compare the externalized and internalizing problems of ASD-HF, ADHD and SLD children using the CBCL questionnaire and comparing them with each other. In a study by Craig et al. [12] was found that children with ADHD reported higher scores in both total and externalizing problems on the CBCL questionnaire than other groups of children with neurodevelopmental disorders, including ASD-HF and SLD. Compared with typically developing subjects, children with these disorders showed higher levels of externalizing, internalizing and behavioral problems mostly in social withdrawal and anxiety/depression [12].

In addition to the intellectual and behavioral profile in children with neurodevelopmental disorders, the stress level of the caregiver was also explored. Parental stress is defined as the aversive psychological reaction to the request to be a parent, typically when the request to be a parent is not associated with the perception of a parent’s available resource [21]. Studies have shown that child, parent, family, and ecological characteristics reciprocally influence each other and determine parental stress [22,23]. These factors are reflected in the Abidin Parental Stress Index (PSI), designed to measure various parental stressors [24]. High levels of parenting stress can negatively impact the general well-being of the family and parents. Previous studies in the literature [12,25] showed that parents of children with ADHD, ASD-HF and SLD report higher parental stress scores than TD children. Potential patient characteristics that may contribute to increased parental stress are emotional problems and cognitive dysfunction.

Since there are very few studies in the literature that have systematically analyzed subjects ADHD, ASD-HF and SLD, the purpose of our study was to trace the neuropsychological profiles of these subjects, evaluate their cognitive, emotional-behavioral functioning and parental stress and compare them with each other and with a control group.

## 2. Materials and Methods

### 2.1. Participants

Our clinical sample consisted of 62 children and adolescents diagnosed with ASD-HF (*n* = 19; males = 13; mean age 8.84 ± 2.36) ADHD (*n* = 21; males = 18; mean age 9.09 ± 1.99) or SLD mixed-type (*n* = 22; males = 14 mean age = 9.77 ± 1.63). All the participants were consecutively recruited to the Child and Adolescents Neuropsychiatry Unit—University-Hospital of Salerno (Italy) after receiving the clinical diagnosis. The diagnoses were made independently by two neuropsychiatrist experts according to DSM-5 criteria. Specifically, ADOS-2 and ADI-R tests were used for the diagnosis of ASD; Conners’ Parent Rating Scale—Revised and Conners’ Teacher Rating Scale—Revised were used for the diagnosis of ADHD; MT 3 clinical tests, Battery for the Evaluation of Dyslexia and Evolutionary Dysorthography-2 (DDE 2), Standardized assessment of calculation and problem-solving skills (ACMT), Battery for the Assessment of Writing and Spelling Proficiency—2 (BVSCO) were used for the diagnosis of SLD.

The control group (TD) consisted of 20 typical children (males = 11) with an average age of 10.88 ± 1.67. The control group was recruited among healthy subjects participating in a screening project on learning difficulties, in which ADHD, ASD and SLD were excluded.

The exclusion criteria for the study were the Total Intelligence Quotient score (TIQ), <70 and the presence of comorbidities and familiarity for neurological (cerebral palsy, epilepsy), psychiatric (anxiety, depression, and psychosis) and other relevant medical conditions. Furthermore, subjects with ADHD, ASD and SLD subjects did not have any comorbidities between them.

All participants carried out a neuropsychological assessment using standardized tests for the assessment of cognitive profile, emotional behavior, and parental stress, as in our clinical practice.

All the subjects recruited agreed to participate in our study. The parents of all the participants provided their written informed consent after receiving a full description about the purpose and the protocol of the study. The study design was approved by the Campania Sud Ethics Committee and it was conducted according to the rules of good clinical practice, in line with the Declaration of Helsinki.

### 2.2. Measures

The neuropsychological assessment included the administration of a direct test to the children for the evaluation of cognitive profile and of two self-administration questionnaires to the parents for the evaluation of the emotional-behavioral problems and the parental stress. Cognitive development was assessed by the Wechsler Intelligence Scale for Children (WISC-IV; Wechsler, 2003) [26]. The WISC-IV provides, in addition to Total Intelligence Quotient score (TIQ), four different indices: Verbal Comprehension Index; Perceptual Reasoning Index; Working Memory Index; lastly Processing Speed Index. A score between 70 and 84 indicates a limit intellectual functioning, while from 85 it is in an average range. The four indices and the Full-Scale IQ are expressed as age-weighted scores, with a mean =100 and a standard deviation =15. The CBCL/6-18 is an evidence-based instrument [27] for evaluating emotional, social, and behavioral problems and functioning in children between the ages of 6 and 18 years. The questionnaire contains 113 items and there are three types of responses recorded on a Likert scale: 0 Not True, 1 Somewhat or Sometimes True, 2 Very True or Often True. The results are divided into many subscales in the form of T scores. According to CBCL normative data, a t-score ≤64 indicates non-clinical symptoms, a t-score between 65 and 69 indicates a borderline range, and a t-score ≥70 indicates clinical symptoms. For the “internalization”, “externalization” and “total” problems subscales, a t-score ≤59 indicates non-clinical symptoms, a t-score between 60 and 64 indicates that the child is in a border range and a t-score ≥65 indicates high levels of maladaptive behavior. The PSI Short Form (PSI/SF) derives by Parenting Complete test of the stress index (PSI) [24] and consists of 36 items for parents of children up to 12 years. Each item requires the parent/guardian whether he agrees, on a five-point Likert scale from strongly agree to strongly disagree, with the statement they read.

In this self-report tool, there are various subscales: Parental Distress (PD), Parent-Child Dysfunctional Interaction Scale (P-CDI), Difficult Child Scale (DC) which respectively evaluate: the level of distress a caregiver is experiencing in his or her parental role, also taking into account personal factors directly related to this role; then how satisfied they are in the relationship with their own child, and lastly how difficult the child is perceived as being [28]. In the PSI/SF, a higher score suggests a higher stress level and a score above 85 indicates (at the 90th percentile) clinically significant parental stress [24]. The total stress score (TS) is obtained by adding the scores of the three subscales PD, PCD-I and DC. The test also includes a defensive response scale (DF) to check the validity of the protocol as it indicates whether the parent tends, for example, to give a better self-image or to minimize problems and perceived stress in the relationship with the child.

### 2.3. Statistical Analysis

First, the raw scores obtained from each subscale of CBCL, and PSI/SF have been transformed into T scores so that an individual’s response can be compared to that of the population norms. In the case of WISC instead, the raw scores were converted into weighted scores. All the scores obtained from the neuropsychological tests were expressed as mean and standard deviation. For the statistical analysis, a comparison was first made between the means of the three groups (ASD-HF, ADHD, SLD) using the non-parametric statistic methods (Kruskall–Wallis H test from which significant differences emerged. To evaluate significant differences between groups, post hoc analysis was conducted using U Mann–Whitney test. P values less than or equal to 0.05 were considered statistically significant.

All data were analyzed using the Statistical Package for Social Science, version 23.0 (IBM Corp. Released 2015. IBM SPSS Statistics for Windows, Version 23.0, IBM Corp: Armonk, NY, USA).

## 3. Results

All the results are summarized in Table 1. In this study, WISC-IV assessment showed in ASD-HF group an average TIQ = 110.16 ± 10.17; they obtained the highest performance in the PRI (118.31 ± 9.55) and the lowest in the PSI (92.05 ± 6.64). There is an average performance in the WMI (101.16 ± 12.84) and a high score in the VCI. The ADHD group obtained an average TIQ = 93.67 ± 9.43; the strength was PRI (96.1 ± 12.6), while the weakness was the PSI (91.90 ± 9.06). VCI (94.43 ± 13.71) and WMI (93.19 ± 10.36) did not different each other. The SLD group obtained an average TIQ score of 96.90 ± 7.14, average performance in the VCI (100.38 ± 10.19), PRI (102.57 ± 8.77) and PSI (95.09 ± 12.39), while a lower performance was recorded, somewhat discrepant with respect to the others, in the working memory index (88.14 ± 9.53). The comparison analysis between clinical groups and TD group, showed significant differences in all WISC-IV indices: Total TIQ (*p* < 0.001), VCI (*p* < 0.001), PRI (*p* < 0.001), WMI (*p* < 0.001), PSI (*p* < 0.001). Post-Hoc analysis revealed that: ADHD group and SLD group had significantly lower scores in TIQ compared to the control group. The VCI index is significantly lower in the ADHD group and significantly higher in the ASD-HF group than in the control group, while there is no significant difference between SLD and control group. The PRI index of the ASD-HF group was significantly higher than TD. No significant differences were detected between SLD group and ADHD group compared to control group in PRI. The WMI is significantly lower in the ADHD group (*p* = 0.001) and in the SLD group (*p* < 0.001) than in the control group. All the clinical groups performed significantly lower than the TD in the PSI. The comparison between the three clinical groups showed that the TIQ score of ASD-HF was significantly higher than SLD (*p* < 0.001) and ADHD groups (*p* < 0.001), that not significantly differed each other (*p* = 0.150). The VCI is significantly higher in ASD-HF group than both SLD (*p* = 0.001) and ADHD groups (*p* < 0.001), while there were not significantly differences between the SLD and ADHD groups (*p* = 0.069). Furthermore, the performance in the PRI is higher in the ASD-HF group than in children with SLD (*p* < 0.001) and with ADHD (*p* < 0.001), while there were no significant differences between the SLD and ADHD groups (*p* = 0.133). The ASD-HF subjects recorded a significantly higher performance in the compared to SLD subjects (*p* = 0.002) while the other comparisons did not give significant results. The comparison between clinical groups and TD group showed significant differences in all the CBCL indices. The post-hoc analysis showed the ASD-HF, ADHD, SLD groups had significantly higher scores compared to the TD group in all the CBCL scales. Regarding the comparison between the three clinical groups, significant differences emerged in three areas: socialization, mood disorders area and somatic disorders area. From the analysis of the PSI/SF scores it emerged that there are significant differences between clinical groups and TD group in all subscales except in the DR subscale of the mothers (*p* = 0.78) (Table 1). There are significant differences only in the DC subscale of mothers between the SLD and ADHD group (*p* = 0.008): the mothers of SLD show a higher score. This section may be divided by subheadings. It should provide a concise and precise description of the experimental results, their interpretation, as well as the experimental conclusions that can be drawn.

## 4. Discussion

The goal of our study was to evaluate the cognitive profiles, emotional/behavioral problems and parental stress in children and adolescents with ASD-HF, ADHD and SLD by comparing the three clinical groups with each other and with a control group.

To the best of our knowledge, there are no previous studies that parallel or compare the neuropsychological profiles of ASD-HF, ADHD and SLD, while also analyzing emotional/behavioral problems and parental stress levels in all three groups.

The results of our study contributed to delineate some neuropsychological characteristics of the three groups of neurodevelopmental disorders.

The ASD-HF group was characterized by a cognitive profile in the normal range with better performances in visual perceptual reasoning skills and in the verbal area. Children and adolescents with ASD-HF appeared to have greater difficulty in rapidly scanning visual stimuli, and in focusing attention. About the emotional-behavioral profile, these subjects experienced both internalizing and externalizing problems; in particular, they could experience problems in social relationships, mood disorders like anxiety and depression, and attention/hyperactivity problems. Both mothers and fathers of these children and adolescents reported high levels of stress in their role as a parent. In addition, there is a high perception of having a difficult child; it is possible that the parents have difficulty obtaining the cooperation of the child or to manage his behavior.

Our study also showed that the neuropsychological profile of children and adolescents with ADHD was characterized by a cognitive level within the norm; although the intellectual functioning was overall homogeneous in the various abilities, the visual-perceptive and verbal abilities were strengths and the processing speed and working memory were weaknesses, thus showing greater difficulty in executive functions in individuals with this neurodevelopmental disorder. As for the emotional aspects, in our study, ADHD children manifested both externalizing and internalizing problems, such as mood disorders, attention problems as well as anxiety and depression problems. Furthermore, parents of ADHD children had high levels of stress and mothers seemed to have more problems interacting with their children.

Finally, the SLD children in our study also showed a peculiar neuropsychological profile. Regarding the intellectual profile, children, and adolescents with SLD showed better performances in verbal and visual-perceptual skills, while the weakest point was working memory. The emotional profile of children with SLD was characterized by internalizing and externalizing problems, such as anxiety, depression, and attention problems. Parents of children with this diagnosis showed high stress levels.

From the comparative analysis of the cognitive abilities, we found that the global intelligence, represented by the Total Intelligence Quotient (TIQ), was in normal range in all three clinical groups. Furthermore, the ADHD and SLD group scored significantly lower compared to ASD-HF group and control group, without significant differences between them [3]. Analyzing more in detail the single sub-indices of the intelligence, we found that Verbal comprehension skills (VCI) were significantly higher in ASD-HF group than all other groups. In Rabiee’s study [3], ASD-HFs subjects showed good Verbal skills (VCI) and Processing Speed abilities (PSI), as well as the control group; in our study, on the other hand, the ASD-HF group scored higher than the TD group in the verbal skills (VCI), but lower in the Processing Speed abilities (PSI), consistent with the results of the study by Oliveras Rentas [10]. As for Visual Perceptual Reasoning (PRI) skills, the ASD-HF group performed significantly better than the other three groups. We can therefore consider this ability the strength of ASD-HF subjects, as confirmed in the literature [3]. Conversely, the ASD-HF group obtains a lower performance in the Processing Speed skill (PSI) than the control group, resulting as the point of weakest in this group. The Working Memory Index (WMI) was significantly lower in SLD and ADHD groups [7,9,13] while Processing Speed Index (PSI) was lower in all three clinical groups compared to controls. Furthermore, we found that all three clinical groups have a significantly different profile and a Cognitive Processing Index (WMI + PSI) lower than the General Ability Index (VCI + PRI) compared to typically developing children. These data agree with the literature [6,9,28] and highlight that children with neurodevelopmental disorders have specific deficiencies related to working memory and processing speed while typical children have a more uniform cognitive profile. These data confirmed a deficit of executive functions, such as focused attention, working memory and graphic-motor skills in all three neurodevelopmental disorders analyzed, compared with typical children [29,30]. In line with our results, the study of Zhang et al. (2020) [31] highlighted that ADHD and ASD-HF subjects performed worse in WMI than typical developing subjects. Another study [13] showed that subjects with SLD go better than subjects with ADHD in executive functions.

Regarding the emotional behavioral profile, all three groups showed externalizing (aggressive behavior, violation of rules) and internalizing (mood and anxiety disorders) problems compared to the control group. The comparison between the clinical groups, showed that the subjects with SLD had less problems in socialization, mood, and somatic disorders than the ASD-HFs and the ADHDs. The data about somatic problems is not in agreement with the data in the literature [12]; our result could be due to an early diagnosis and to an early treatment of subjects with SLD which may had allowed them to not develop somatic symptoms.

Concerning PSI/SF, the Total Stress of parents of all the three clinical groups was higher than those of the parents of typically developing children. The Parental Distress (PD) score was higher in ASD-HF and ADHD group compared to controls, revealing a higher perception of stress related to parental role in these two groups. The perception of having a Difficult Child (DC) was higher in all three groups than in the control, particularly mothers in all three clinical groups had a higher perception of having a difficult child than fathers. On the Difficult Child (DC) scale, mothers of children with SLD scored significantly higher than ADHD. One possible explanation might be that school performance is important for mothers and they lose more confidence in the child’s ability to achieve good academic competence [12]. Compared to the control group, the perception of having a complicated relationship with children (P-CDI) was significantly higher in all parents despite the heterogeneity of the disorders. Our results about parental stress agree with those of Craig et al. (2016) [12]. Parents of children who received diagnosis of neurodevelopmental disorder, experience higher levels of stress than parents of typically developing children. These results lead to consider necessary a possible support intervention also for the parents of children with SLD, who generally are not taken into consideration in this aspect, and to improve the quality of family life, especially in children who also have emotional behavioral problems.

The results of our study may be useful to better understand the characteristics and specificities of the neurodevelopmental disorders considered, and to support a precise differential diagnosis. These findings can also clarify the strengths and weaknesses of the children and adolescents receiving these diagnoses. Knowing more precisely the main characteristics and differences of neurodevelopmental disorders can be of great help to clinicians working in the sector to identify and propose early and targeted treatments. For example, children with falls in specific cognitive dimensions could benefit from early treatment on those areas. Treating children with specific disorders early could prevent emotional-behavioral symptoms (such as anxiety or depression or low self-esteem) that could affect their quality of life. Finally, underlining the presence of stress in the parents of children with neurodevelopmental disorders, allows us to understand that it is important to take care of the whole family unit to allow a harmonious development of the child. There are very few studies comparing children and adolescents with ASD, ADHD and LSD, for this reason our study could bring more information on the possible presence of specific differences between these groups.

Some limitations inherent this study should be reported. The first limitation is related to the stress assessment procedure used. Although PSI and CBCL have good psychometric properties and are fundamental when assessing internalized states, they are subjective self-related measures that could lead to feasible prejudices. Our work is a cross-sectional study, in the future it will be useful to carry out prospective studies to evaluate the development trajectory. Furthermore, adaptive functioning could also be considered, and the executive functions of the three groups of patients could be assessed with specific tools. Moreover, another limitation of this study is the sample size. It will certainly be appropriate to expand the sample and carry out more complex statistical analyzes.

## 5. Conclusions

Our study highlighted that the three neurodevelopmental disorders considered (Autism Spectrum Disorder level 1 ASD-HF, Attention Deficit/Hyperactivity Disorder—ADHD and Specific Learning Disorder—SLD) had a peculiar neuropsychological cognitive profile that distinguishes itself from the others and characterizes them in their functioning. The global intelligence was within the normal range in all the three groups, although ASD-HF scored higher than ADHD and SLD. The SLD subjects have weaknesses in working memory and in the processing speed skills. The ASD-HF had a strength in logical reasoning and a weakness in the processing speed (hand-eye coordination). ADHD subjects had a weakness in the verbal comprehension, working memory and processing speed abilities. Despite the heterogeneity of three the clinical conditions, the emotional-behavioral problems were very present in all groups compared to the controls, with greater problems of socialization, mood, and somatic problems in the ASD-HF. Compared to total stress, all the parents of ASD-HF, ADHD and SLD showed higher levels of stress than parents of typically developing children, despite the different clinical condition.

## Figures and Tables

**Table 1 brainsci-11-00584-t001:** Statistical comparison between the average scores of the different groups.

	SLD (m ± SD)	ADHD (m ± SD)	ASD-HF (m ± SD)	Control (m ± SD)	Kruskal–Wallis H Test		U Mann–Witney Post-Hoc Test					
							SLD vs. Control	ADHD vs. Control	ASD-HF vs. Control	SLD vs. ADHD	SLD vs. ASD-HF	ASD-HF vs. ADHD
**WISC-IV**												
TIQ	96.90 ± 7.14	93.67 ± 9.43	110.16 ± 10.17	104.60 ± 8.30	30.97	0.000 *	*p* = 0.003 *	*p* = 0.000 *	*p* = 0.052	*p* = 0.150	*p* = 0.000 *	*p* = 0.000 *
VCI	100.38 ± 10.19	94.43 ± 13.71	111.00 ± 10.27	103.15 ± 8.46	19.66	0.000 *	*p* = 0.233	*p* = 0.009 *	*p* = 0.021 *	*p* = 0.069	*p* = 0.001 *	*p* = 0.000 *
PRI	102.57 ± 8.77	96.1 ± 12.6	118.31 ± 9.55	104.35 ± 14.42	25.42	0.000 *	*p* = 0.771	*p* = 0.126	*p* = 0.003 *	*p* = 0.133	*p* = 0.000 *	*p* = 0.000 *
WMI	88.14 ± 9.53	93.19 ± 10.36	101.16 ± 12.84	103.30 ± 8.42	22.80	0.000 *	*p* = 0.000 *	*p* = 0.001 *	*p* = 0.396	*p* = 0.090	*p* = 0.002 *	*p* = 0.082
PSI	95.09 ± 12.39	91.90 ± 9.06	92.05 ± 6.64	103.00 ± 5.84	19.59	0.000 *	*p* = 0.024 *	*p* = 0.000 *	*p* = 0.000 *	*p* = 0.367	*p* = 0.365	*p* = 0.765
**CBCL**												
Total problems	60.09 ± 10.54	63.76 ± 11.57	64.31 ± 9.52	51.50 ± 5.45	20.33	0.000 *	*p* = 0.002 *	*p* = 0.000 *	*p* = 0.000 *	*p* = 0.224	*p* = 0.218	*p* = 0.914
Anxiety/Depression	62 ± 11.92	65.57 ± 9.57	65.16 ± 9.09	54.40 ± 5.67	20.72	0.000 *	*p* = 0.023 *	*p* = 0.000 *	*p* = 0.000 *	*p* = 0.067	*p* = 0.156	*p* = 0.694
Withdrawn/Depressed	60.95 ± 11.79	62.14 ± 9.22	62.84 ± 10.35	51.80 ± 4.82	23.19	0.000 *	*p* = 0.000 *	*p* = 0.000 *	*p* = 0.000 *	*p* = 0.367	*p* = 0.364	*p* = 0.838
Somatic complaints	58.81 ± 9.08	61.76 ± 8.43	64.05 ± 10.29	54.25 ± 5.40	13.45	0.004 *	*p* = 0.157	*p* = 0.002 *	*p* = 0.001 *	*p* = 0.148	*p* = 0.071	*p* = 0.586
Social problems	60.90 ± 10.59	62.57 ± 11.69	67.05 ± 10.59	54.80 ± 4.09	15.04	0.002 *	*p* = 0.073	*p* = 0.019 *	*p* = 0.000 *	*p* = 0.549	*p* = 0.032 *	*p* = 0.134
Thought problems	54.85 ± 5.75	59.66 ± 10.12	59.58 ± 9.97	53.55 ± 4.95	11.15	0.011 *	*p* = 0.274	*p* = 0.008 *	*p* = 0.004 *	*p* = 0.099	*p* = 0.060	*p* = 0.838
Attention problems	62.95 ± 11.43	66.09 ± 14.48	66.21 ± 12.42	52.80 ± 5.54	18.02	0.000 *	*p* = 0.000 *	*p* = 0.000 *	*p* = 0.000 *	*p* = 0.464	*p* = 0.282	*p* = 0.754
Rule breaking behavior	56.52 ± 7.20	59.38 ± 9.12	58.74 ± 6.50	50.35 ± 3.82	6.65	0.000 *	*p* = 0.003 *	*p* = 0.000 *	*p* = 0.000 *	*p* = 0.321	*p* = 0.192	*p* = 0.881
Aggressive Behavior	58.95 ± 10.68	62.81 ± 12.49	58.58 ± 9.75	50.603.95	17.18	0.001 *	*p* = 0.002 *	*p* = 0.000 *	*p* = 0.003 *	*p* = 0.283	*p* = 0.000	*p* = 0.293
Affective problems	61.0 ± 8.47	67.14 ± 11.49	66.79 ± 9.41	53.75 ± 4.77	25.54	0.000 *	*p* = 0.003 *	*p* = 0.000 *	*p* = 0.000 *	*p* = 0.110	*p* = 0.039 *	*p* = 0.860
Anxiety problems	62.38 ± 7.46	65.14 ± 8.95	64.05 ± 7.94	54.40 ± 4.31	20.91	0.000 *	*p* = 0.000 *	*p* = 0.000 *	*p* = 0.000 *	*p* = 0.341	*p* = 0.762	*p* = 0.634
Somatic problems	55.09 ± 7.73	60.95 ± 6.52	60.95 ± 7.78	55.50 ± 5.53	15.79	0.001 *	*p* = 0.426	*p* = 0.005 *	*p* = 0.010 *	*p* = 0.003 *	*p* = 0.006 *	*p* = 0.978
Attention deficit/Hyperactivity Problems	60.14 ± 7.10	62.24 ± 9.34	61.58 ± 8.07	54.30 ± 3.83	10.95	0.003 *	*p* = 0.001 *	*p* = 0.005 *	*p* = 0.004 *	*p* = 0.551	*p* = 0.875	*p* = 0.957
Oppositional defiant problems	56.38 ± 5.32	59.62 ± 9.14	57.05 ± 7.29	50.80 ± 3.76	8.32	0.000 *	*p* = 0.000 *	*p* = 0.001 *	*p* = 0.002 *	*p* = 0.538	*p* = 0.833	*p* = 0.428
Conduct problems	56.95 ± 8.54	59.76 ± 9.62	58.16 ± 7.28	49.60 ± 3.76	5.46	0.000 *	*p* = 0.001 *	*p* = 0.000 *	*p* = 0.000 *	*p* = 0.506	*p* = 0.384	*p* = 0.774
Internalizing	61.0 ± 10.38	67.14 ± 11.49	65.47 ± 11.54	53.35 ± 4.78	19.42	0.000 *	*p* = 0.002 *	*p* = 0.000 *	*p* = 0.000 *	*p* = 0.154	*p* = 0.276	*p* = 0.946
Externalizing	55.90 ± 9.59	60.62 ± 11.88	56.58 ± 10.01	49.00 ± 3.37	11.19	0.004 *	*p* = 0.025 *	*p* = 0.001 *	*p* = 0.003 *	*p* = 0.154	*p* = 0.724	*p* = 0.296
**PSI/SF**												
TS mothers	89.05 ± 7.52	85.24 ± 10.89	85.53 ± 10.12	44.75 ± 20.29	39.14	0.000 *	0.000 *	0.000 *	0.000 *	*p* = 0.154	*p* = 0.283	*p* = 0.648
PD mothers	49.28 ± 31.79	62.86 ± 25.77	59.47 ± 20.94	43.00 ± 20.86	11.15	0.011 *	0.073	0.004 *	0.006 *	*p* = 0.119	*p* = 0.416	*p* = 0.384
P-CDI mothers	64.05 ± 23.11	67.38 ± 27.46	68.16 ± 18.27	36.50 ± 25.65	14.86	0.002 *	0.002 *	0.001 *	0.006 *	*p* = 0.971	*p* = 0.378	*p* = 0.550
DC mothers	94.28 ± 7.63	91.43 ± 9.24	90.79 ± 11.09	50.00 ± 19.87	27.27	0.000 *	0.000 *	0.001 *	0.000 *	*p* = 0.008 *	*p* = 0.116	*p* = 0.394
DR mothers	54.05 ± 26.15	52.86 ± 29.18	66.84 ± 21.42	64.50 ± 20.29	6.82	0.078	0.391	0.052	0.038	*p* = 0.145	*p* = 0.101	*p* = 0.653
TS fathers	84.76 ± 9.42	88.33 ± 9.40	85.53 ± 15.26	45.25 ± 16.36	43.76	0.000 *	0.000 *	0.000 *	0.000 *	*p* = 0.411	*p* = 0.268	*p* = 0.730
PD fathers	55.09 ± 22.17	67.38 ± 27.46	62.89 ± 21.69	39.00 ± 20.43	10.40	0.015 *	0.273	0.005 *	0.002 *	*p* = 0.164	*p* = 0.454	*p* = 0.282
P-CDI fathers	66.90 ± 29.73	60.24 ± 24.92	62.37 ± 23.71	31.00 ± 21.98	21.14	0.000 *	0.000 *	0.002 *	0.000 *	*p* = 0.567	*p* = 0.906	*p* = 0.586
DC fathers	60.24 ± 24.92	76.67 ± 25.17	80.79 ± 25.94	55.75 ± 21.96	38.62	0.000 *	0.000 *	0.000 *	0.000 *	*p* = 0.309	*p* = 0.439	*p* = 0.922
DR fathers	60.24 ± 24.92	67.86 ± 26.81	71.84 ± 26.15	67.00 ± 20.86	12.55	0.006 *	0.017 *	0.249	0.000 *	*p* = 0.826	*p* = 0.072	*p* = 0.225

m = mean; SD = Standard Deviation; SLD = Specific Learning Disorders; ADHD = Attention Deficit/Hyperactivity Disorder; ASD-HF = Autism Spectrum Disorder level 1/High functioning; TIQ = Total Intelligence Quotient; VCI = Verbal Comprehension Index; PRI = Perceptual Reasoning Index; WMI = Working Memory Index; PSI = Processing Speed Index; TS = Total Stress; PD = Parental Distress; P-CDI = Parent-Child Dysfunctional Interaction; DR = Defensive Responding; DC = Difficult Child. asterisks (*) mark significant differences.

## Data Availability

The data presented in this study are available on request from the corresponding author.

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
