# Peer review of "Neuropsychological Profile, Emotional/Behavioral Problems, and Parental Stress in Children with Neurodevelopmental Disorders"

_brainsci, 2021, doi:10.3390/brainsci11050584_

Round 1
Reviewer 1 Report
Thank you for the opportunity to review your manuscript. Here are some suggestions for improving your work:
Introduction:
Authors should add a reference after the first sentence, on line 44. And another reference on line46, after the second sentence.
Once you have introduced ASD, SLD, ADHD suggested that you be consistent and use it throughout the text, for example, ASD on line 51.
You are missing a period on line 59 before Furthermore,
Line 66 when you indicate WISC-IV you should previously introduce it and answer what it has to do in your study, even if it is obvious to psychologists, think that the manuscript can be read by other professionals who may not be so familiar with the test and its usefulness. On the other hand, I think it would be interesting to talk more about cognitive processes rather than tests per se, in the introduction.
Line 103, suggested to the authors that they change CBCL test to CBCL questionnaire.
On the other hand, the authors barely address the issue of parental stress in the introduction. At least one paragraph should be introduced to it in order to understand the importance of it and to justify the relationship and why the study was conducted.
Materials and Methods:
Line 134 should indicate that TIQ refers to the total IQ score of the WISC-IV (which is introduced later). You should also indicate where and how the neurotypical group was recruited.
You should also indicate the type of sampling, whether it was purposive, probabilistic,...etc. and then comment on whether or not this is a limitation of the results.
You should indicate whether the study was approved by an ethics committee and whether consent to participate was obtained from the legal guardians.
In statistical analysis, you should also indicate the software and version that was used to do the same.
Results:
I suggest that you modify the presentation of results according to the neurocognitive processes you want to address and not by tests or subtests.
You need to give the correct format to Table 1. according to the publication norms.
I think could be done other types of more complex analyses.
The discussion should be rephrased on the previous changes.
Author Response
REWIEWER 1
Introduction:
- Authors should add a reference after the first sentence, on line 44. And another reference on line 46, after the second sentence.
Author response: - We thank the reviewer for the comment. We have added the references, as suggested.
- Once you have introduced ASD, SLD, ADHD suggested that you be consistent and use it throughout the text, for example, ASD on line 51.
Author response: - We thank the reviewer for the comment. We made the required changes.
- You are missing a period on line 59 before Furthermore,
Author response: - We thank the reviewer for the comment. The missing period has been added.
- Line 66 when you indicate WISC-IV you should previously introduce it and answer what it has to do in your study, even if it is obvious to psychologists, think that the manuscript can be read by other professionals who may not be so familiar with the test and its usefulness. On the other hand, I think it would be interesting to talk more about cognitive processes rather than tests per se, in the introduction.
Author response: - We thank the reviewer for the suggestion. We have added more information regarding the WISC-IV test.
- Line 103, suggested to the authors that they change CBCL test to CBCL questionnaire.
Author response: - We thank the reviewer for the suggestion. We made this change, as suggested.
- On the other hand, the authors barely address the issue of parental stress in the introduction. At least one paragraph should be introduced to it in order to understand the importance of it and to justify the relationship and why the study was conducted
Author response: - We thank the reviewer for the suggestion. We have better introduced information about parental stress in the introduction section: “In addition to the intellectual and behavioral profile in children with neurodevelopmental disorders, the stress level of the caregiver was also explored. Parental stress is defined as the aversive psychological reaction to the request to be a parent, typically when the request to be a parent is not associated with the perception of a parent's available resource [21]. Studies have shown that child, parent, family, and ecological characteristics reciprocally influence each other and determine parental stress [22, 23]. These factors are reflected in the Abidin Parental Stress Index (PSI), designed to measure various parental stressors [24]. High levels of parenting stress can negatively impact on the general well-being of the family and parents. Previous studies in the literature [12,25] showed that parents of children with ADHD, ASD-HF and SLD report higher parental stress scores than TD children. Potential patient characteristics that may contribute to increased parental stress are emotional problems and cognitive dysfunction.”
Materials and Methods:
- Line 134 should indicate that TIQ refers to the total IQ score of the WISC-IV (which is introduced later). You should also indicate where and how the neurotypical group was recruited.
Author response: - We thank the reviewer for the suggestion. We changed the reference TIQ and we have added the required information on the Neurotypical Group: “The control group was recruited among healthy subjects participating in a screening project on learning difficulties, in which ADHD, ASD and SLD were excluded.”
- You should also indicate the type of sampling, whether it was purposive, probabilistic,...etc. and then comment on whether or not this is a limitation of the results.
Author response: - We thank the reviewer for the suggestion. We proceeded to add information on the type of sampling: “All the participants were consecutively recruited to the Child and Adolescents Neuropsychiatry Unit- University-Hospital of Salerno (Italy) after receiving the clinical diagnosis. […] All the subjects recruited agreed to participate in our study.”
- You should indicate whether the study was approved by an ethics committee and whether consent to participate was obtained from the legal guardians.
Author response: - We thank the reviewer for the suggestion. We have included this point, as follow: “The parents of all the participants provided their written informed consent after receiving a full description about the purpose and the protocol of the study. The study design was approved by the Campania Sud Ethics Committee and it was conducted according to the rules of good clinical practice, in line with the Declaration of Helsinki.”
- In statistical analysis, you should also indicate the software and version that was used to do the same.
Author response: - We thank the reviewer for the suggestion. We added the required information, as follow: “All data were analyzed using the Statistical Package for Social Science, version 23.0 (IBM Corp. Released 2015. IBM SPSS Statistics for Windows, Version 23.0, IBM Corp: Armonk, NY, USA).”
Results:
- I suggest that you modify the presentation of results according to the neurocognitive processes you want to address and not by tests or subtests.
Author response: - We thank the reviewer for the suggestion. We made these changes as suggested.
- You need to give the correct format to Table 1. according to the publication norms.
Author response: - We thank the reviewer for the suggestion. We changed the format of Table 1
- I think could be done other types of more complex analyses.
Author response: - We thank the reviewer for the suggestion. We will certainly carry out more complex analyzes in the next studies. We have added this point to the limitations and future perspectives of the study: “It will certainly be appropriate to expand the sample and carry out more complex statistical analyzes.”
Discussion:
- The discussion should be rephrased on the previous changes.
Author response: - We thank the reviewer for the comments and suggestions. We rephrased the discussion section, adding substantial parts in the discussion section, as required: “The results of our study contributed to delineate some neuropsychological characteristics of the three groups of neurodevelopmental disorders. The ASD-HF group was characterized by a cognitive profile in the normal range with better performances in visual perceptual reasoning skills and in the verbal area. Children and adolescents with ASD-HF appeared to have greater difficulty in rapidly scanning visual stimuli, and in focusing attention. About the emotional-behavioral profile, these subjects experienced both internalizing and externalizing problems; in particular, they could experience problems in social relationships, mood disorders like anxiety and depression, and attention/hyperactivity problems. Both mothers and fathers of these children and adolescents reported high levels of stress in their role as a parent. In addition, there is a high perception of having a difficult child; it is possible that the parents have difficulty obtaining the cooperation of the child or to manage his behavior.
Our study also showed that the neuropsychological profile of children and adolescents with ADHD was characterized by a cognitive level within the norm; although the intellectual functioning was overall homogeneous in the various abilities, the visual-perceptive and verbal abilities were strengths and the processing speed and working memory were weaknesses, thus showing greater difficulty in executive functions in individuals with this neurodevelopmental disorder. As for the emotional aspects, in our study, ADHD children manifested both externalizing and internalizing problems, such as mood disorders, attention problems as well as anxiety and depression problems. Furthermore, parents of ADHD children had high levels of stress and mothers seemed to have more problems interacting with their children.
Finally, also the SLD children in our study showed a peculiar neuropsychological profile. Regarding the intellectual profile, children, and adolescents with SLD showed better performances in verbal and visual-perceptual skills, while the weakest point was working memory. The emotional profile of children with SLD was characterized by internalizing and externalizing problems, such as anxiety, depression, and attention problems. Parents of children with this diagnosis showed high stress levels.”

Reviewer 2 Report
This study tries to compare three different groups of individuals with neurodevelopmental disabilities using some used widely applied scales. The aim is cited at the end of the introduction part. I was expecting to see a theoretical framework for this study in this part and I think this is a serious shortcoming with this study.
I was expecting that “the Neuropsychological profile” as the main aim of this study to be explained. What I noticed in this study is comparing three groups of individuals with neurodevelopmental disabilities to be compared with each other and a control group, I did not understand if the used scales might be considered as a Neuropsychological profile.
How did the sample receive information about the study (word of mouth, pamphlet. Leaflet, poster, website advertisement or any other possible way)? How many parent and families approached how many accepted? Regarding the exclusive criteria how many potential participants excluded?
I also notice the application of words such as “patient” to refer to this group of individuals (in the abstract) and in other parts. The question is that because the sample was recruited in a hospital the members besides their neuropsychological diagnosis also had other illness? Or they called patients because they received neuropsychological diagnosis! (The second possibility makes the situation more difficult and problematic with the sample). This issue needs to be addressed. Application of “patient” for individuals with ASD and SLD is not applicable in this field.
I also think that the following sentence needs a revision (pare 3 -line 117):
“searches showed that” you mean searching in the databases or library searches or possibly “research finding?”
Also page 9 line 66 needs revision:
“Our results about parental stress are il keeping with those of Craig et al. (2016) [11].”
More information is needed regarding the diagnosis of these 3 groups of individuals. I understand that DSM criteria might also be applicable for a clinical diagnosis but for the research poupous application of the diagnostic instruments is recommended (ADI-R or ADOS for ASD, Conners scale for ADHD or other diagnostic scales for SLD).
I also recommend adding some comments regarding the readership of these findings. Which group of stakeholders in this field might benefit from the findings? What is the application of this study?
You missed a section on the limitation of your study. Sample size, recruitment and also problem with the diagnosis all worth it to be mentioned in this part.
Author Response
REVIEWER 2
- This study tries to compare three different groups of individuals with neurodevelopmental disabilities using some used widely applied scales. The aim is cited at the end of the introduction part. I was expecting to see a theoretical framework for this study in this part and I think this is a serious shortcoming with this study.
Author response: We thank the reviewer for the comments and suggestions. Additional information on the theoretical part has been added, as follow:
“To diagnose ADHD, symptoms must be present in two different contexts, develop before age 12, and have a negative impact on psychosocial functioning [1]. DSM-5 includes in neurodevelopmental disorders also the Specific Learning Disorders (SLD), in which individuals have an impairment of reading, writing, and calculating skills, despite a normal intelligence. Deficient academic skills are far below the expected level for the individual's age and involve interference with academic performance and/or daily living activities. Difficulties are not best explained by intellectual disability, impaired vision or hearing, lack of knowledge of the language of school instruction or inadequate education [1].
One of the most frequently used tests to assess intelligence in school-aged individuals is the Wechsler Intelligence Scale for Children – Fourth Edition (WISC-IV).
In addition to the intellectual and behavioral profile in children with neurodevelopmental disorders, the stress level of the caregiver was also explored. Parental stress is defined as the aversive psychological reaction to the request to be a parent, typically when the request to be a parent is not associated with the perception of a parent's available resource [21]. Studies have shown that child, parent, family, and ecological characteristics reciprocally influence each other and determine parental stress [22, 23]. These factors are reflected in the Abidin Parental Stress Index (PSI), designed to measure various parental stressors [24]. High levels of parenting stress can negatively impact on the general well-being of the family and parents. Previous studies in the literature [12,25] showed that parents of children with ADHD, ASD-HF and SLD report higher parental stress scores than TD children. Potential patient characteristics that may contribute to increased parental stress are emotional problems and cognitive dysfunction.”
- I was expecting that “the Neuropsychological profile” as the main aim of this study to be explained. What I noticed in this study is comparing three groups of individuals with neurodevelopmental disabilities to be compared with each other and a control group, I did not understand if the used scales might be considered as a Neuropsychological profile.
Author response: We thank the reviewer for the comments and suggestions. We understand that the definition of a neuropsychological profile may be unclear, but neuropsychologists working with children commonly infer a CNS basis for many developmental disabilities based on test performance and behavioral observations. The intrinsic characteristics of neuropsychological methods have more to do with the manner in which certain test results or behavioral observations are interpreted. The conviction that these results can be used to infer a CNS basis for a given developmental disability, more than any other single factor, is the distinctive characteristic of neuropsychological investigation. Based on this conviction, the neuropsychologist attempts to discover abilities that reflect cerebral status as opposed to other factors and use these abilities to determine CNS contributions to childhood disorders (Jack M. Fletcher a & H. Gerry Taylor, “Neuropsychological approaches to children: Towards a developmental neuropsychology”, 2015, Journal of Clinical Neuropsychology).
- How did the sample receive information about the study (word of mouth, pamphlet. Leaflet, poster, website advertisement or any other possible way)? How many parent and families approached how many accepted? Regarding the exclusive criteria how many potential participants excluded?
Author response: We thank the reviewer for the comments. The information you requested has been added: “All the participants were consecutively recruited to the Child and Adolescents Neuropsychiatry Unit- University-Hospital of Salerno (Italy) after receiving the clinical diagnosis.”
- I also notice the application of words such as “patient” to refer to this group of individuals (in the abstract) and in other parts. The question is that because the sample was recruited in a hospital the members besides their neuropsychological diagnosis also had other illness? Or they called patients because they received neuropsychological diagnosis! (The second possibility makes the situation more difficult and problematic with the sample). This issue needs to be addressed. Application of “patient” for individuals with ASD and SLD is not applicable in this field.
Author response: We thank the reviewer for pointing out these important details. Children with neurodevelopmental disorders were recruited after receiving the neuropsychiatric diagnosis. All children with a specific neurodevelopmental disorder had no other comorbid diseases or disorders.
- I also think that the following sentence needs a revision (pare 3 -line 117): “searches showed that” you mean searching in the databases or library searches or possibly “research finding?”
Author response: We thank the reviewer for the comments. The period has been made clearer.
- Also page 9 line 66 needs revision:“Our results about parental stress are il keeping with those of Craig et al. (2016) [11].”
Author response: We thank the reviewer for the comments. The error has been corrected.
- More information is needed regarding the diagnosis of these 3 groups of individuals. I understand that DSM criteria might also be applicable for a clinical diagnosis but for the research poupous application of the diagnostic instruments is recommended (ADI-R or ADOS for ASD, Conners scale for ADHD or other diagnostic scales for SLD).
Author response: We thank the reviewer for the comments. All tests used for diagnosis have been added to the article, as follow: “Specifying, ADOS-2 and ADI-R tests were used for the diagnosis of ASD; Conners’ Parent Rating Scale – Revised and Conners’ Teacher Rating Scale - Revised were used for the diagnosis of ADHD; MT 3 clinical tests, Battery for the Evaluation of Dyslexia and Evolutionary Dysorthography-2 (DDE 2), Standardized assessment of calculation and problem-solving skills (ACMT), Battery for the Assessment of Writing and Spelling Proficiency – 2 (BVSCO) were used for the diagnosis of SLD.”
- I also recommend adding some comments regarding the readership of these findings. Which group of stakeholders in this field might benefit from the findings? What is the application of this study?
Author response: We thank the reviewer for the comments. We added this information to the article, as follow: “The results of our study may be useful to better understand the characteristics and specificities of the neurodevelopmental disorders considered, and to support a precise differential diagnosis. These findings can also clarify the strengths and weaknesses of the children and adolescents receiving these diagnoses. Knowing more precisely the main characteristics and differences of neurodevelopmental disorders can be of great help to clinicians working in the sector to identify and propose early and targeted treatments. For example, children with falls in specific cognitive dimensions could benefit from early treatment on those areas. Treating children with specific disorders early could prevent emotional-behavioral symptoms (such as anxiety or depression or low self-esteem) that could affect their quality of life. Finally, underlining the presence of stress in the parents of children with neurodevelopmental disorders, allows us to understand that it is important to take care of the whole family unit to allow a harmonious development of the child. There are very few studies comparing children and adolescents with ASD, ADHD and LSD, for this reason our study could bring more information on the possible presence of specific differences between these groups.”
- You missed a section on the limitation of your study. Sample size, recruitment and also problem with the diagnosis all worth it to be mentioned in this part.
Author response: We thank the reviewer for the comments. We added limitation in discussion section, as follow: “Some limitations inherent this study should be reported. The first limitation is related to the stress assessment procedure used. Although PSI and CBCL have good psychometric properties and are fundamental when assessing internalized states, they are subjective self-related measures that could lead to feasible prejudices. Our work is a cross-sectional study, in the future it will be useful to carry out prospective studies to evaluate the development trajectory. Furthermore, adaptive functioning could also be considered, and the executive functions of the three groups of patients could be assessed with specific tools. Moreover, another limitation of this study is the sample size. It will certainly be appropriate to expand the sample and carry out more complex statistical analyzes.”

Round 2
Reviewer 2 Report
This version is an updated form of the previously submitted manuscript.
The authors have done all their best to addressed the raised issues. There is one point which I want to stress. I appreciate that the authors considered substituting "patient" with some other more appropriate term, hence in the added information I noticed the application of "patient" in two separate parts:
- Line 140, page 3
- Line 112, page 12